# A bed load transport equation based on the spatial distribution of shear stress - Oak Creek revisit

Angel Monsalve[1], Catalina Segura[2], Nicole Hucke[1], Scott Katz[2,3]

[1] Departamento de Ingeniería de Obras Civiles, Universidad de la Frontera, Francisco Salazar 01145, Temuco, Chile.
[2] Forest Engineering, Resources, and Management, Oregon State University, 201 Peavy Hall, Corvallis, OR 97331, USA
[3] Natural Systems Design, Bellingham WA. 98225

*Correspondence to*: Angel Monsalve (angel.monsalve@ufrontera.cl)

**Abstract:** Bed load transport formulations for gravel bed-rivers are often based on reach-averaged shear stress values. However, the complexity of the flow field in these systems results in wide distributions of shear stress, whose effects on bed load transport are not well captured by the frequently used bed load transport equations, leading to inaccurate estimates of sediment transport. Here, we modified a subsurface-based bed load transport equation to include the complete distributions of shear stress generated by a given flow within a reach. The equation was calibrated and verified using bed load data measured at Oak Creek, OR. The spatially variable flow field characterization was obtained using a two-dimensional flow model calibrated over a wide range of flows between 0.1 and 1.0 of bankfull discharge. The shape of the distributions of shear stress was remarkably similar across different discharge levels which allowed it to be parameterized in terms of discharge using a Gamma function. When discharge is high enough to mobilize the pavement layer (1.0 m$^3$/s in Oak Creek), the proposed transport equation had a similar performance to the original formulation based on reach-averaged shear stress values. In addition, the proposed equation predicts bed load transport rates for lower flows when the pavement layer is still present because it accounts for bed load transport occurring in a small fraction of the channel bed that experience high values of shear stress. This is an improvement over the original equation, which fails to estimate this bed load flux by relying solely on reach-average shear stress values.

## 1 Introduction

Predicting bed load is both expensive and practically challenging, as data from a wide range of flows is required to develop robust relationships between discharge and load. In addition, characterizing bed load at high flow levels—that transport the majority of the sediment is often dangerous (Bunte et al., 2008). Samples collected using hand-held devices can be widely variable due to factors related to variations of their orifice size and the sampling time (Beschta, 1981; Emmett, 1980; Pitlick, 1988; Vericat et al., 2006). While advances in safe, accurate sediment sampling technology such as bed load traps (Bunte et al., 2008), radio tracers (Bradley and Tucker, 2012; May and Pryor, 2014; Olinde and Johnson, 2015; Schmidt and Ergenzinger, 1992), and acoustic impact methods (Rickenmann and McArdell, 2007; Turowski and Rickenmann, 2011; Wyss et al., 2016a, 2016b, 2016c; Yager et al., 2012b) provide possible alternatives to hand-held samplers, field efforts remain expensive and out of reach for many practical applications.

Bed load modeling can be a convenient alternative to measuring bed load in the field. The development of empirical bed load relationships has progressed significantly over the past three decades such that many formulations allow for the estimation of bed load based on hydraulic and grain size information. In general sediment transport equations are based on reach-averaged one-dimensional shear stress estimates and the surface (e.g., Barry et al., 2004; Parker, 1990; Recking, 2013; Wilcock and Crowe, 2003) or subsurface (Parker et al., 1982; Parker and Klingeman, 1982) grain size information.

Many of these sediment transport equations (e.g., Parker et al., 1982; Parker and Klingeman, 1982) were developed based on data from Oak Creek, a steep, coarse, gravel-bed stream in the Oregon Coast Range (Milhous, 1973). The Oak Creek dataset was collected using a vortex sampler between 1969–1990; data from 1971 was published in the thesis work of Milhous (1973). The dataset is unique because the vortex sampling method enabled to capture the entire bed load flux of sand–cobble size particles for a wide range of flows over long time periods, reducing the error associated with hand-held samplers (Parker et al., 1982). Although it has been reported that the efficiency of the vortex sampler decreased for smaller grain sizes (Milhous, 1973; O'leary and Beschta, 1981), the Oak Creek dataset remains one of the most comprehensive to date. The Oak Creek based transport equations were developed by collapsing the relations between reference conditions for the motion of different grain sizes into single functions (i.e., a similarity collapse) (Einstein, 1950; Parker, 1990; Parker et al., 1982; Parker and Klingeman, 1982). Both Parker and Klingeman (1982) and Parker et al. (1982) limited their analysis to flows during which the surface channel layer was mobilized ("pavement" was broken) to develop their transport functions. Parker et al., (1982) computes total bed load ($Q_b$) based on a single grain size (the median – $D_{50}$) whereas Parker and Klingeman (1982) expands that relationship to the entire grain size distribution (GSD). This is accomplished by introducing a hiding function that accounts for differences in the hiding and exposure of particles to the flow in mixed-sized beds. Additionally, Parker and Klingeman (1982) incorporated a low flow transport relation to estimate the GSD of $Q_b$ at a full range of flows. Later, Parker (1990) modified Parker and Klingeman (1982) equations to be based in the surface grain size distribution.

Although the transport relations of Parker et al. (1982) and Parker and Klingeman (1982) have been successfully applied to many rivers, the work of Recking (2013b) highlighted the variability that can be incorporated into $Q_b$ estimates due to uncertainty in input shear stress ($\tau$) values. The high spatial variability in $\tau$ throughout a river reach has been well documented (Clayton and Pitlick, 2007; Katz et al., 2018; Lisle et al., 2000; May et al., 2009; McDonald et al., 2010; Monsalve et al., 2016; Recking, 2013a; Segura and Pitlick, 2015; Yager et al., 2018). However, most transport functions, including Parker and Klingeman (1982) and Parker et al., (1982), utilized reach-averaged estimates of $\tau$ in their calculations and are highly sensitive to uncertainties in these values due to the non-linear exponents on each function (Recking, 2013a). Significant differences in bed load estimates computed using $\tau$ from one- (1D) and two-dimensional (2D) approximations have been found because of the spatial variability of $\tau$ (Ferguson, 2003; Gomez and Church, 1989; Recking, 2013a). Thus, the simplification of $\tau$ to a 1D variable may not capture spatial changes in bed load associated with localized values of high $\tau$ (Segura and Pitlick, 2015).

The objective of this study is to develop a bed load transport equation based on the subsurface GSD that uses the complete shear stress distribution for different discharge levels within a specific reach. Although a surface-based equation (e.g., Parker, 1990) would have been stronger at estimating bed load transport at low flows, we adopted a subsurface-based equation (Parker and Klingeman, 1982) because it has fewer parameters to adjust. A subsurface-based equation also allows considering sand sizes, which are commonly found in the bed load (Clayton and Pitlick,

2007; Hassan and Church, 2001; Lisle, 1995; Mueller et al., 2005; Recking, 2010; Segura and Pitlick, 2015). This new approach is developed using field measurements of bed load transport rates and GSD, river topography, and 2D flow modeling. The performance of the new equation is then tested using the historic Oak Creek dataset (Milhous, 1973). Specific objectives of our study are to:

        i)      Analyze the characteristics of shear stress distributions over a wide range of discharge levels

ii)     Generate synthetic shear stress distributions based solely on discharge

      iii)    Modify a reach-averaged subsurface-based equation (Parker and Klingeman, 1982) developed for Oak Creek to use complete shear stress distributions

      iv)    Test the performance of the proposed equation for a wide range of discharge level

## 2 Methods

### 2.1 Study area

This study was conducted in Oak Creek, a cobble-gravel stream located in the Oregon Coast Range (Milhous, 1973, figures 1 and 2). The catchment drains 7 km$^2$ of forest land underlain by basaltic lithology (Milhous, 1973; O'Connor et al., 2014). The climate is Mediterranean with wet winters and cool/mild summers. Elevations within the Oak Creek

watershed range from 143 to 664 m (Paustian and Beschta, 1979). The basin is located in the McDonald-Dunn Forest, which is owned and managed by the College of Forestry at Oregon State University and dominated by Douglas fir (*Pseudotsuga menziesii*) and Oregon white oak (*Quercu sp*). In the riparian area, vegetation is dominated by alder (*Alnus sp*), black cottonwood (*Papulus trichocarpa*), and big leaf maple (*Acer macrophyllus*) with lower densities of Douglas fir and Oregon white oak. The 150-m study reach has a pool-riffle sequence in the upstream end and a

relatively straight section in the downstream section (Katz et al., 2018) and is located directly upstream from a historic sediment transport sampling facility where bed load samples were collected between 1969 and 1973 (Milhous, 1973). The site has a rectangular cement weir in which a stage-discharge ($Q$) relationship has been developed (Katz et al, 2018). The stream has a slope ($S_b$) of 0.014 m/m and bankfull dimensions of 6 m in width and 0.46 m in depth. Recent field observations indicate that bankfull discharge ($Q_{bf}$) is 3.4 m$^3$/s (Katz et al, 2018), which is similar to the bankfull

discharge reported almost 40 years ago by Milhous (1973). The stream bed is armored with coarser surface overlying a finer subsurface. The surface $D_{50}$ is 45 mm while the subsurface $D_{50_s}$ is 21 mm (Katz et al., 2018) (figure 2)

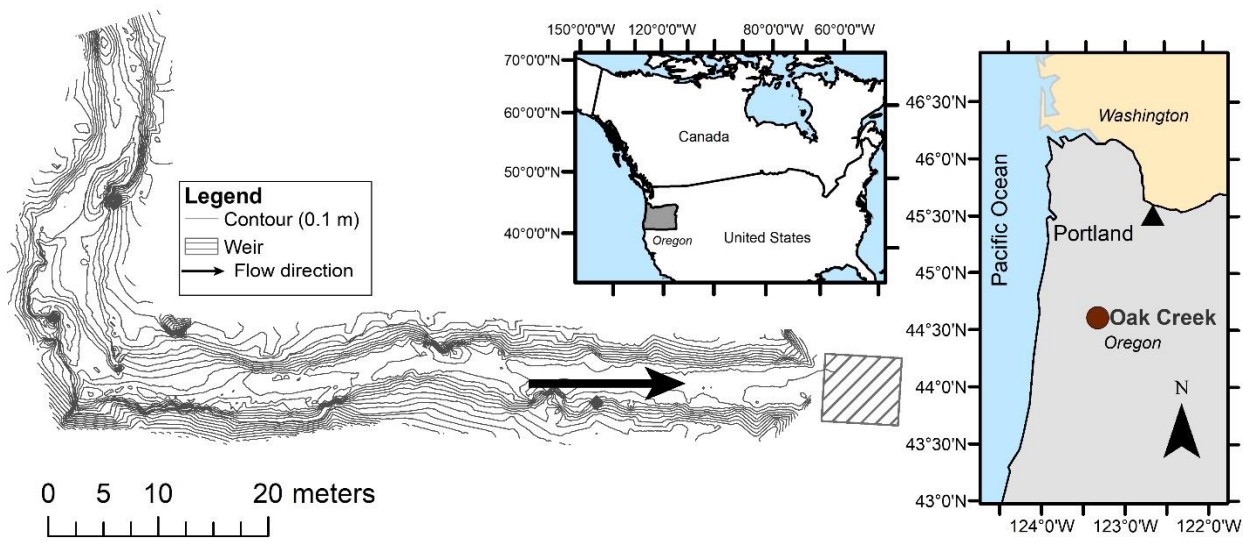

Figure 1: Location of the study reach in Oak Creek, Oregon (44° 23' 19.092'' N, 123° 19' 51.312' W). Contours every 0.1 meters are indicated.

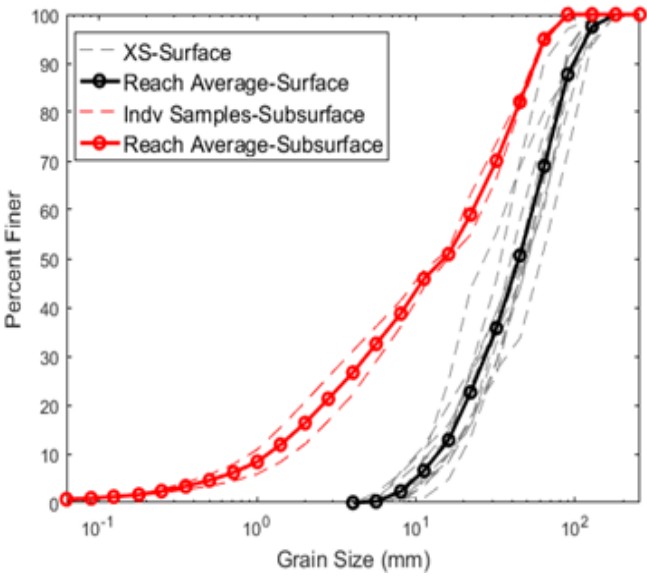

Figure 2: Surface and subsurface grain size distribution (GSD). The average surface GSD is based on 23 cross-sections (XS) and average subsurface GSD based on 2 samples of the substrate collected from exposed bars**.**

## 2.2 Two-dimensional modeling

Spatial distributions of the flow field, in particular local shear stresses, were estimated for seven discharges (0.4 to 3.4

120   m³/s, equivalent to 0.12 $Q_{bf}$ to $Q_{bf}$) using the Flow and Sediment Transport with Morphological Evolution of Channels (FaSTMECH) two-dimensional flow solver (McDonald et al., 2010). Specific details of the modeling effort can be found in Katz et al. (2018). The model has also been described and used in several studies (e.g., Clayton and Pitlick, 2007; Conner and Tonina, 2014; Kinzel et al., 2009; Lisle et al., 2000; Maturana et al., 2014; Mcdonald et al., 2005; Monsalve et al., 2016; Mueller and Pitlick, 2014; Nelson and McDonald, 1995; Nelson and Smith, 1989; Nelson

et al., 2010; Segura and Pitlick, 2015), therefore, only the most relevant characteristics of it are described here. The model uses a finite difference solution to the vertically integrated conservation of mass and momentum equations (Nelson et al., 2003) with calculations performed in an orthogonal curvilinear grid that follows the surveyed planform topography of the channel (Nelson and Smith, 1989). Roughness is included using a unitless drag coefficient ($C_d$). A zero-equation model for the lateral eddy viscosity (*LEV*) that assumes homogeneous and isotropic turbulence is used for turbulence closure (Barton et al., 2005; Miller and Cluer, 1998; Nelson et al., 2003). For our models $C_d$ ranged from 0.017 to 0.04 and *LEV* ranged 0.0010 to 0.0032 (Katz et al., 2018). The calibration indicated strong model fits in terms of water surface elevation with root mean square errors (RMSE) between 0.025 and 0.048 m and $R^2 > 0.99$ (Katz et al., 2018).

The local shear stress ($\tau_{xy}$) was calculated at every grid node in the model domain as a function of $C_d$, the vertically averaged streamwise ($u$) and cross-stream ($v$) velocities, and water density ($\rho$), assumed as 1,000 kg/m$^3$.

$$\tau_{xy} = \rho C_d \left( u_{xy}^2 + v_{xy}^2 \right) \tag{Eq. 1}$$

where the subscripts $x$ and $y$ correspond to the stream-wise and cross-stream directions.

### 2.3 Shear stress distribution analysis

Characteristics of the distributions of predicted $\tau_{xy}$ were analyzed as a function of discharge. We produced histograms of the mean-normalized shear stress distribution ($\tau/\langle\tau\rangle$) (subscripts *x* and *y* were dropped for simplicity) to compare patterns between flows. For each flow level we fitted the frequency distributions of $\tau/\langle\tau\rangle$ to a two-parameter Gamma function (Nicholas, 2000; Paola, 1996; Pitlick et al., 2012; Recking, 2013a; Segura and Pitlick, 2015):

$$f(\tau) = \frac{\alpha^\alpha (\tau/\langle\tau\rangle)^{(\alpha-1)} e^{-\alpha(\tau/\langle\tau\rangle)}}{\langle\tau\rangle \Gamma(\alpha)} \tag{Eq. 2}$$

where $\Gamma$ is the standard Gamma function, $\alpha$ is the shape parameter and $\beta = \langle\tau\rangle/\alpha$ is the scale parameter. The parameters of the Gamma function that best fitted the distributions were found using the maximum likelihood estimation (MLE) method (Bevington and Robinson, 2003). We assessed the goodness of fit of the Gamma function in each flow event by computing the room mean square error (RMSE) and the reduced chi-square ($\chi_v^2$), defined as chi-square ($\chi^2$) divided by the number of degrees of freedom, according to:

$$\chi^2 = \sum \frac{[f_k - f(x_k)]^2}{\sigma_k^2} \tag{Eq. 3}$$

where and $f_k$ and $f(x_k)$ are observed and predicted mean-normalized shear stress frequencies in a given bin interval, $k$. The uncertainty associated with the observed frequencies, $\sigma_k^2$, was estimated as the square of the number of observations in each bin (Bevington and Robinson, 2003; Press et al., 2007). Initially, in all cases, we specified the bin width using the Freedman–Diaconis rule (Freedman and Diaconis, 1981). To improve statistics when the number of $\tau$ values in a given bin was less than five we joined two consecutive bins until all bins had five or more $\tau$ values. Typically, for the used goodness of fit indicators, an excellent fit is $\chi_v^2 \leq 1$ and RMSE of zero (Bevington and Robinson, 2003; Press et al., 2007).

## 2.4 Sediment transport equations

The original subsurface-based sediment transport equation of Parker and Klingeman (1982) was modified to explicitly consider the spatial distribution of shear stress. This equation was chosen because it was developed from measurements collected in the same reach as this study, it gives accurate estimates of bed load transport, and it is relatively simple to extend for our purposes (see below). The modified version of the Parker and Klingeman (1982) equation was formulated such that it accounts for the bed load transported by each increment of shear stress, which means that it considered the range of local contributions of $\tau$ across the channel bed. By doing so, all $\tau$ values, even those less-frequent high-magnitude shear stresses, are explicitly included in the calculations. To obtain the new equation the parameters of the Einstein bed load function ($G$) proposed by Parker (1978) were relaxed and fitted as new parameters. The parameter values were optimized based on the fit of volumetric transport rate per unit width of channel ($q$) and the bed load GSD. Like the original equation, we only consider discharges of approximately 1 m³/s or higher to calibrate the new equations (for calibration purposes, our lower discharge was 0.99 m³/s). The fitting procedure of the parameters minimized the absolute error between predicted and measured $q$ and maximized the Nash-Sutcliffe efficiency index (Nash and Sutcliffe, 1970) using the calculated and observed bed load GSD. Equal importance (equal weight) was given to the fit of $q$ and to the fit of the bed load GSD.

The new equation was based on the locally dimensionless shear stress ($\tau^*$):

$$\tau^* = \frac{\tau}{(\rho_s - \rho)gD} \tag{Eq. 4}$$

where $\rho_s$ is the sediment density, $g$ is the acceleration due to gravity, and $D$ the grain size. Notice that for a given flow discharge $\tau^*$ has a distribution of values depending on the local $\tau$ (previously defined as $\tau_{xy}$) and variations in the fraction of the GSD. The original transport relation of Parker and Klingeman (1982) (equation 5) is valid for uniform grain sizes and $\phi > 1$, with $\phi$ being the transport stage (equation 6)

$$G = \frac{W^*}{W_r^*} = 5.6 \cdot 10^{-3} \left(1 - \frac{0.853}{\phi}\right)^{4.5} \tag{Eq. 5}$$

where the subscript $r$ denotes a reference value associated with a small but measurable transport rate. Transport stage ($\phi$) is defined as:

$$\phi = \tau^*/\tau_r^* \tag{Eq. 6}$$

The dimensionless transport rate, $W^*$ (equation 5) is defined as:

$$W^* = \frac{(s-1)gq}{(\tau/\rho)^{1.5}} \tag{Eq. 7}$$

where $s$ is the specific gravity of sediment ($s = \rho_s/\rho$)

We extended equation 5 to include all grain size fractions in the subsurface GSD ($D_i$, subscript $i$ denotes the size range) and $\phi_i > 0.95$. In the most general form the equation for the dimensionless transport rate is $W_i^* = 0.0025G_i'$,

where the constant is the reference transport rate of Parker and Klingeman (1982) ($W_r^* = 0.0025$) and $G_i'$ is the new (modified) transport relation. The proposed relation is a two-part equation applicable to sediment mixtures:

$$W_i^* = 0.0025 \cdot 10^{-3} exp(26.6(\phi_i - 1) - 19.53(\phi_i - 1)^2) \quad \text{, for} \quad 0.95 < \phi_i < 1.65$$

$$W_i^* = 0.57 \left(1 - \frac{0.853}{\phi_i}\right)^{4.5} \quad \text{, for} \quad \phi_i \geq 1.65$$

(Eq. 8)

To account for the mobility of individual grain sizes we used the Parker and Klingeman (1982) hiding function:

$$\frac{\tau_{ri}^*}{\tau_{r50}^*} = \left(\frac{D_i}{D_{50}}\right)^{-0.982}$$

(Eq. 9)

where $\tau_{r50}^*$ =0.0876 is the reference Shields stress for the median grain size of the subsurface Parker and Klingeman (1982) obtained for the same reach. The transport stage (equation 6), valid for any grain size $D_i$ and for the entire distribution of shear stress values (i.e., every $\tau_i^*$), was re-written as:

$$\phi_i = \tau_i^*/\tau_{ri}^*$$

(Eq. 10)

To obtain the volumetric transport rate the predicted shear stresses were grouped in a series of intervals ($\tau_j$, subscript $j$ denotes an interval of $\tau$ values) with a regular shear stress increment ($\Delta\tau_j = 0.25$ N/m$^2$). For all discharges, $\tau_j$ was defined such that it ranges from zero to the maximum predicted shear stress value. For a given $D_i$ and $\tau_j$ the volumetric transport rate per unit width ($q_{ij}$) is:

$$q_{ij} = \frac{\left(\frac{\tau_j}{\rho}\right)^{1.5} F_i W_{ij}}{(s-1)g} f_{\tau_j}$$

(Eq. 11)

where $F_i$ is the volume fraction of the $i^{th}$ grain-size class in the subsurface GSD, $W_{ij}$ is calculated using equation 8 for each $\tau_j$ and $f_{\tau_j}$ is the fraction of the bed area where a certain $\tau_j$ acts. The width-integrated volumetric transport rate for a given flow event is:

$$q_b = b \sum_i \sum_j q_{ij}$$

(Eq. 12)

with $b$ being the width of the gravel bed. In all bed load estimations sand grains likely to move in suspension were excluded, thus the subsurface GSD was truncated at 2 mm.

## 3 Results

### 3.1 Spatial distributions of shear stress

The numerical models allowed the characterization of the spatial distribution of $\tau$ for each discharge level (figure 3). In terms of reach-averaged values, the predicted $\langle\tau\rangle$ varied between 18.3 and 51.1 N/m$^2$ for flows between 0.12 and 1.0 $Q_{bf}$ (table 1). Furthermore, the mean shear stress, $\langle\tau\rangle$, scaled with discharge such that an exponential function explained 97% of its variance (figure 4 a). The predicted $\langle\tau\rangle$ were 66 to 79 % smaller than the mean shear stress values calculated based on the depth-slope product (table 1).

Table 1: Summary of model shear stress distributions including reach-averaged shear stress ($\langle\tau\rangle$), mean modeled water depth ($h$), and total shear stress ($\tau_t = \rho g h S_b$), measured bed load transport rate ($q_{b\_meas}$) (Milhous, 1973), Gamma fit parameters ($\alpha$ and $\beta$), and goodness of fit: reduced chi-square ($\chi_v^2$) and root mean square error (RMSE) between the observed distribution of shear stress and the gamma fit predicted distribution.

| $Q$ (m$^3$/s) | $\langle\tau\rangle$ (N/m$^2$) | $h$ (m) | $\tau_t$ (N/m$^2$) [a] | $q_{b\_meas}$ (kg/s) | $\alpha$ | $\beta$ | $\chi_v^2$ | RMSE |
|---|---|---|---|---|---|---|---|---|
| 0.40 | 18.34 | 0.18 | 24.72 | 1.17·10-5 | 7.49 | 0.133 | 0.151 | 0.05 |
| 0.64 | 23.10 | 0.22 | 30.32 | 3.65·10-4 | 6.46 | 0.155 | 0.102 | 0.03 |
| 0.99 | 24.64 | 0.25 | 34.73 | 3.01·10-3 | 5.95 | 0.168 | 0.074 | 0.03 |
| 1.33 | 25.60 | 0.28 | 38.32 | 1.51·10-2 | 5.64 | 0.177 | 0.055 | 0.03 |
| 1.46 | 26.16 | 0.29 | 39.74 | 2.00·10-2 | 5.55 | 0.180 | 0.043 | 0.03 |
| 1.91 | 32.76 | 0.34 | 46.52 | 2.8·10-2 | 4.82 | 0.207 | 0.070 | 0.03 |
| 3.40 | 51.12 | 0.47 | 64.40 | 3.78·10-1 | 3.69 | 0.271 | 1.026 | 0.07 |
| a: We estimated the total shear stress ($\tau_t$) assuming uniform flow (depth-slope product) and a constant energy slope of 0.014 m/m. | | | | | | | | |

The shape of the distributions of $\tau/\langle\tau\rangle$ was remarkably similar across all modeled discharges (figure 3). In all cases the highest frequencies of local $\tau$ were around the mean value and approximately 92% of the predicted $\tau/\langle\tau\rangle$ were below 2. We fitted the normalized shear stress distributions to Gamma functions with $\alpha$ parameters that varied between 7.49 and 3.60 and $\beta$ parameters that varied between 0.13 to 0.27 (Table 1). These parameters, $\alpha$ and $\beta$, varied linearly with discharge (figure 4 b). In both cases discharge explained more than 92% of the variability in $\alpha$ and $\beta$.

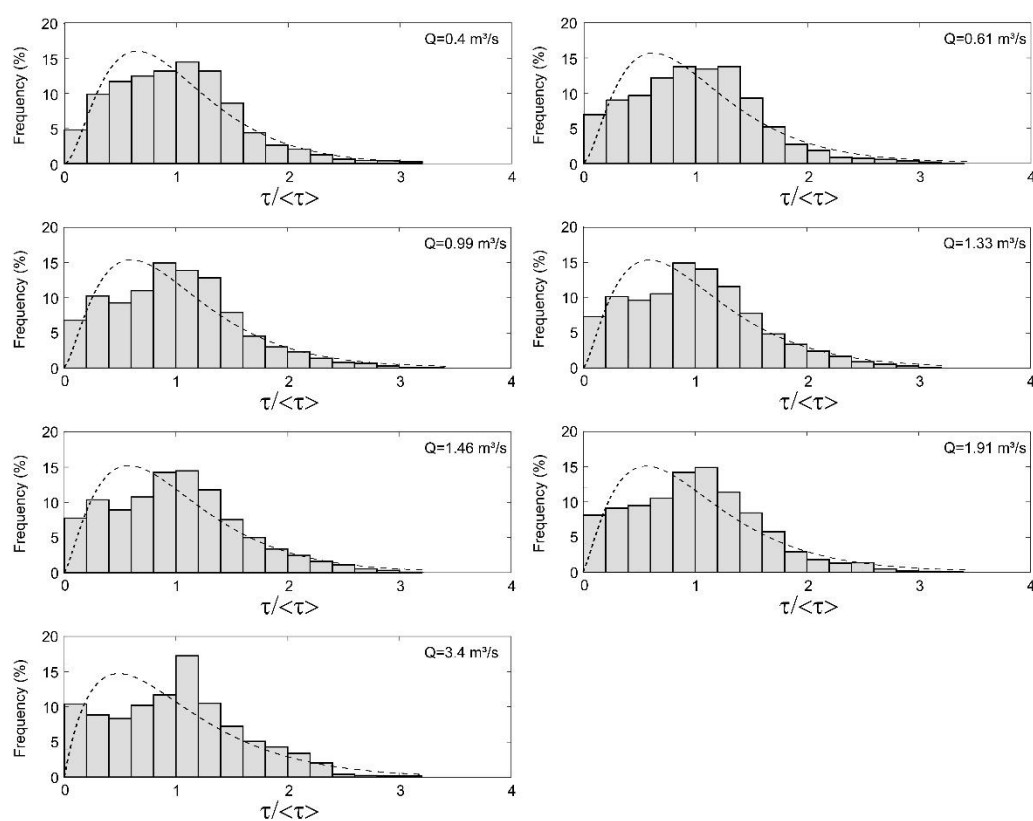

Figure 3: Frequency distributions of mean-normalized shear stress ($\tau/\langle\tau\rangle$) for the seven discharge levels. Fitted Gamma distribution curves are shown as dashed lines. Discharges values are indicated in the upper right corner of each panel.

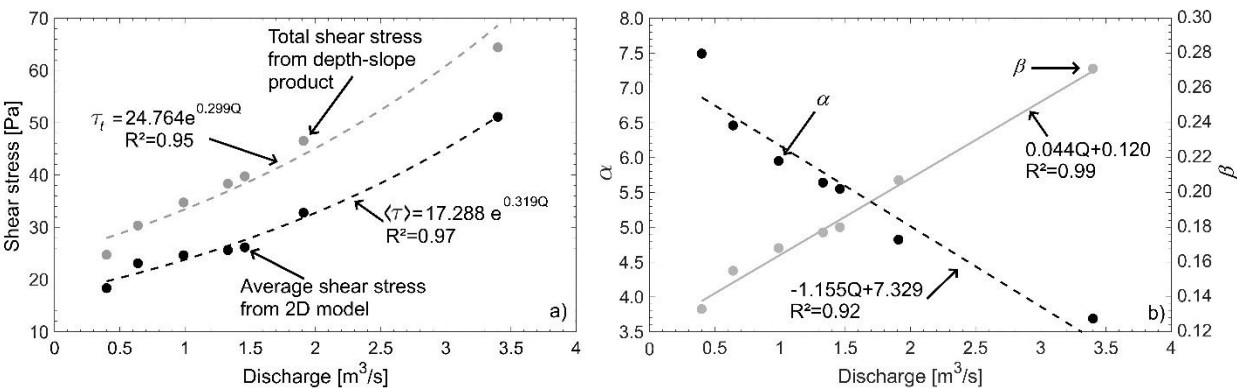

Figure 4: (a) Relationship between the reach-averaged shear stress ($\langle\tau\rangle$), 1D total shear stress ($\tau_t$) and discharge ($Q$). (b) Relationship between the parameters of the Gamma function ($\alpha$ and $\beta$) and discharge. Total shear stress from depth slope product has a similar trend than $\langle\tau\rangle$ but it is consistently higher. All our results are based on the 2D models and $\tau_t$ is shown here only for comparison purposes.

The equations that relate the Gamma fit parameters and the reach-averaged shear stress to the discharge are:

$$\alpha = -1.155Q + 7.329 \tag{Eq. 13}$$

$$\beta = 0.044Q + 0.120 \tag{Eq. 14}$$

$$\langle\tau\rangle = 17.288e^{0.319Q} \tag{Eq. 15}$$

Combining equations equation 13 and 15 with equation 2, an expression to estimate the distribution of $\tau$ for any given $Q$ can be obtained. Additionally, these synthetic distributions can be used to evaluate the accuracy of our bed load transport equation for discharge levels different than those used for its calibration (see section 3.3).

**3.2 Characteristics of our sediment transport relation**

The proposed sediment transport equation has the same shape as the Parker and Klingeman (1982) relation, but it is scaled such that $W_i^*$ is consistently lower for all $\phi_i$ values (figure 5). The consistently lower $W_i^*$ indicates that bed load transport occurs are relatively small localized areas of the bed where stress stresses are higher than the average value. While calibrating this formulation we kept some key features of the original equation in Parker and Klingeman

(1982), thus, we reduced the number of degrees of freedom. Specifically, the shape of both equations is the same and are valid within the same $\phi_i$ intervals. We used an exponential function with a second-degree polynomial function as argument for $0.95 < \phi_i < 1.65$ and a power function with an exponent equal to 4.5 for $\phi_i \geq 1.65$. We also maintained $\tau_{r50}^* = 0.0876$ and the exponent of the hiding function (0.982) as fixed values (i.e., were not adjusted while calibrating our equations).

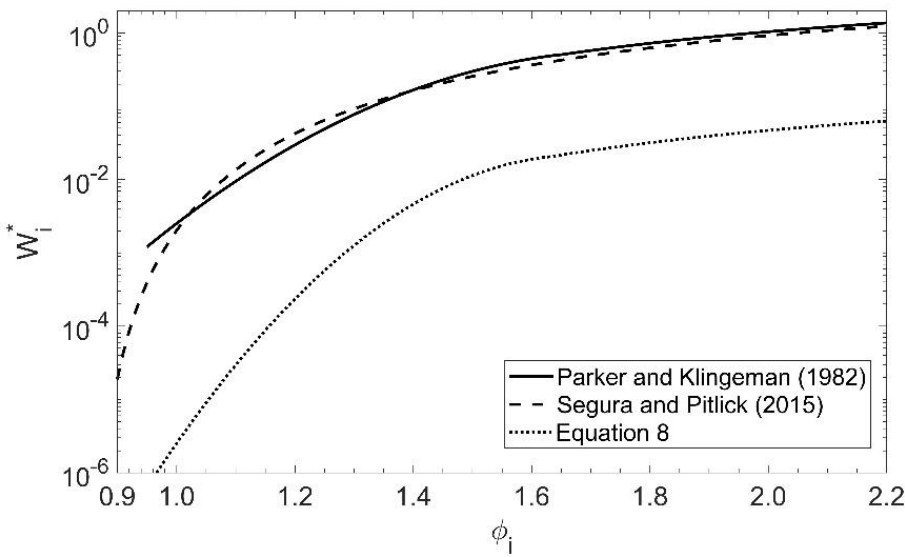

Figure 5: Comparison between different subsurface-based sediment transport equations and the one proposed in this study. The relation of Segura and Pitlick (2015), which is also a modified version of Parker and Klingeman (1982), is shown as reference.

### 3.3 Sediment transport calculations

All flow events used for calibrating had an error of less than an order of magnitude between the measured and predicted bed load transport rate (figure 6, table 2). In terms of sediment transport estimates this order of error is generally considered as a relatively strong estimation (Yager et al., 2007, 2012b). Similarly to the Parker and Klingeman (1982) equation, our bed load estimates for flows lower than 0.4 m³/s were weaker. This is not surprising given that these low flows were not used for calibration and that there are very low rates of transport at such low discharges (~10% of $Q_{bf}$). The equation of Parker and Klingeman (1982) was not designed to include distributions of $\tau$. However, to have a point of comparison we contrasted the measured and predicted bed load rates for the original Parker and Klingeman (1982) equation applied over the complete shear stress distribution predicted by the 2D numerical model instead of our formulation. While our estimated bed load rates for $Q \geq 0.99$ m³/s were within one order of magnitude of the observed value, those predicted using the Parker et al. (1982) equation were consistently over estimated by over an order of magnitude in all cases (figure 6).

In terms of prediction of the bed load GSD, the Nash-Sutcliffe efficiency index was in all cases greater than 0.65. (Table 2). For $Q \geq 1.33$ m³/s the efficiency index was greater or equal to 0.85 (Table 2). The difference between predicted and observed bed load median grain sizes ($D_{50_{meas}} - D_{50_{pred}}$) was lower than 10 mm in all these cases. For the $Q = 0.99$ m³/s event the error in the median grain size was larger (12.3 mm) with predicted grain size values consistently coarser (figure 7).

Equation 8 is only applicable when the spatial distribution of $\tau$ is known. However, this is not the case in most studies and practical applications. In our case, given the strong correlations between discharge and reach-averaged shear stress and also between discharge and the Gamma function parameters, combining equations 13 and 15 with equation 2

allowed us to generate synthetic distributions of $\tau$ for a given flow of interest. We tested the accuracy of our equation
when these synthetic distributions were used as input using a subset of the Milhous (1973) database (grey circles in
figure 6). The scenarios considered correspond to the same 22 flow events used by Parker and Klingeman (1982) in
their analysis and had flow discharges that ranged between 1.02 and 3.4 m$^3$/s. Using the synthetic distributions of $\tau$,
our equation predicted bed load rates within an order magnitude of error for all 22 events. Considering the logarithm
of the ratio between the measured and predicted bed load transport rate ($log\big(q_{b\_pred}/q_{b\_meas}\big)$, which is a measure of
the accuracy of an estimation (0 indicates perfect agreement and ±1 an error of an order of magnitude, Yager et al.,
2007), the estimated bed load rates had a median $log\big(q_{b\_pred}/q_{b\_meas}\big)$ of -0.07, minimum of -0.84, 25$^{th}$ percentile
of -0.42, 75$^{th}$ percentile of 0.17, and a maximum of 0.55.

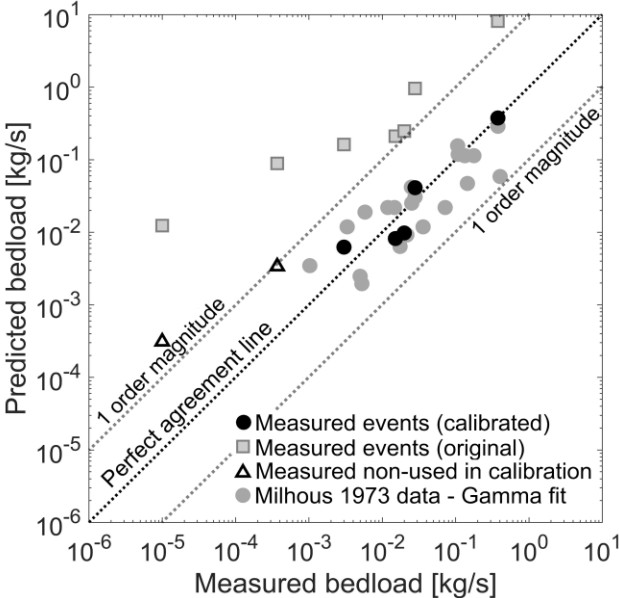

Figure 6: Comparison between measured and predicted bed load transport rate for different methods and data sets.
Five events ($Q \geq$0.99 m$^3$/s) were used when calibrating equation 8 (black circles). Triangles represent the estimated
bed load using equation 8 for two low flow events ($Q < 0.64$ m$^3$/s) that were not used for calibration The equation
of Parker and Klingeman (1982) applied locally to the complete shear stress distributions is shown as reference
(squares). Additionally, a synthetic spatial shear distribution based on equation 2 and the parameters given in
equations 13–15 was used with our equation to calculate the bed load rate (grey circles). Measured field data were
collected by Milhous (1973).

Table 2: Modeled ($q_{b\_pred}$) and observed ($q_{b\_meas}$) bed load transport rates and modeled ($D_{50\_pred}$), Nash-Sutcliffe
efficiency index, and observed ($D_{50\_meas}$) median grain size for the events used in the calibration of equation 8.

| $Q$ (m$^3$/s) | $q_{b\_meas}$ (kg/s) | $q_{b\_pred}$ (kg/s) | $log\left(\dfrac{q_{b\_pred}}{q_{b\_meas}}\right)$ | Nash-Sutcliffe efficiency index | $D_{50\_meas}$ (mm) | $D_{50\_pred}$ (mm) |
|---|---|---|---|---|---|---|
| 0.99 | $3.01\cdot10^{-3}$ | $6.23\cdot10^{-3}$ | 0.32 | 0.66 | 7.7 | 20.0 |
| 1.33 | $1.51\cdot10^{-2}$ | $0.82\cdot10^{-2}$ | -0.27 | 0.94 | 13.6 | 19.9 |
| 1.46 | $2.00\cdot10^{-2}$ | $0.98\cdot10^{-2}$ | -0.31 | 0.85 | 10.2 | 20.1 |
| 1.91 | $2.8\cdot10^{-2}$ | $4.11\cdot10^{-2}$ | 0.17 | 0.97 | 18.8 | 21.3 |
| 3.40 | $3.78\cdot10^{-1}$ | $3.76\cdot10^{-1}$ | 0.00 | 0.88 | 30.1 | 22.5 |

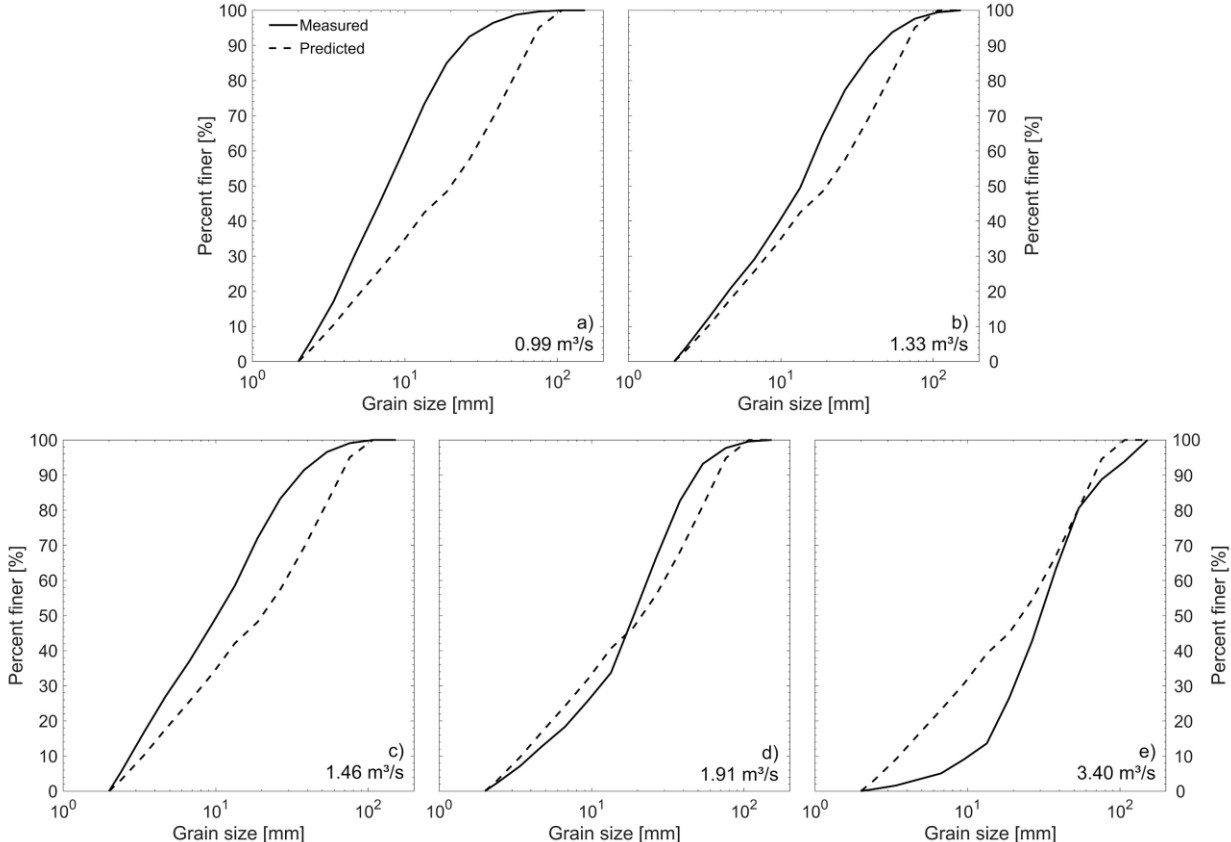

Figure 7: Measured and predicted bed load grain size distributions for all events used in the calibration of equation 8. Flow discharges are shown in the lower-left corner of each panel.

Based on the local shear stress we identified the areas of the bed where most of the bed load likely occurs for a given flow level (figure 8). Similar to the study of Segura and Pitlick (2015), in terms of bed load transport rate per unit width, the size of these areas increases with discharge. At 0.29 $Q_{bf}$ ($Q$ =0.99 m³/s) most bed load transport occurs in a relatively small, localized area of the bed, occupying approximately 5.4% of the total bed area (100% is the wetted area under bankfull flow conditions). The percentage increases to 7.6% at 0.43 $Q_{bf}$ ($Q$ =1.46 m³/s) and 17.5% for 0.56 $Q_{bf}$ ($Q$ =1.91 m³/s). At bankfull flow conditions ($Q$ =3.40 m³/s) the proportion of the channel bed that is predicted to be mobile is 52.5%, and mainly concentrated along the thalweg. It is important to mention that this method provides only an approximation to the region where most bed load occurs. In this approach we are not considering bed evolution and other time-dependent processes that may alter the location of areas where bed load transport occurs.

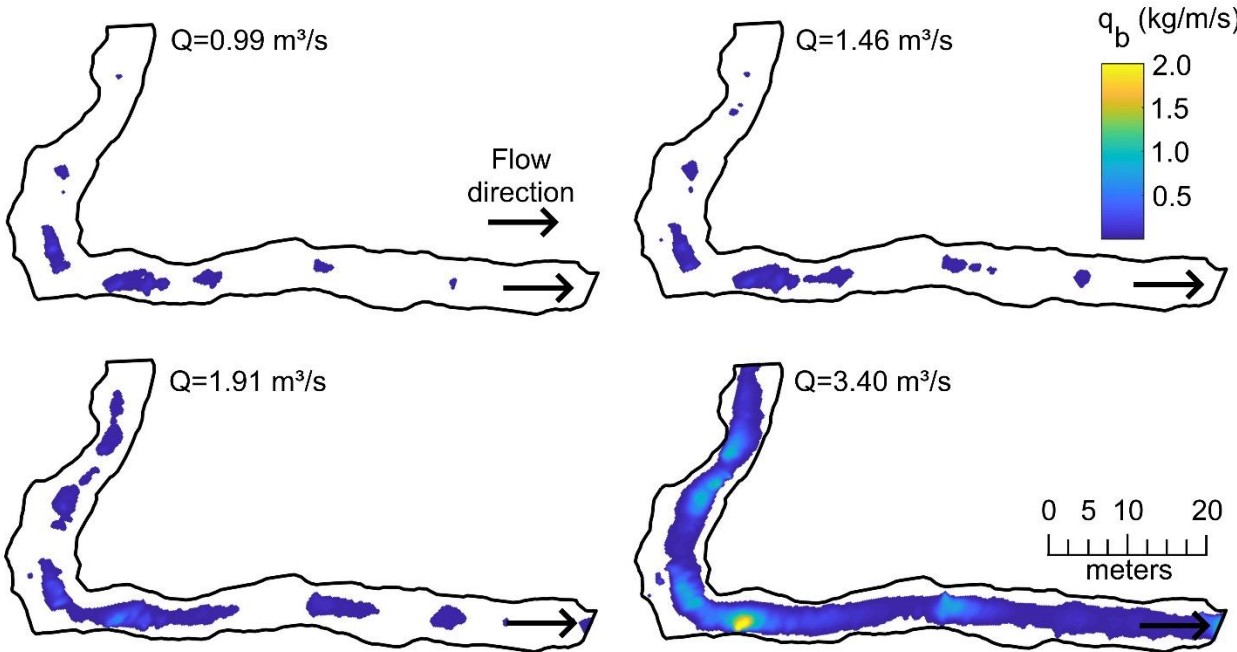

Figure 8: Predicted local bed load transport rate per unit width $(q_b)$ for four flows levels between 0.29 to 1.00 $Q_{bf}$. White areas in each case indicate zero predicted transport. Black lines correspond to the wetted area under bankfull flow conditions. Maps of the predicted local shear stress for the same study area and flow scenarios are available in Katz et al. (2018)

## 4 Discussion

### 4.1 Using spatial distribution of shear stress to estimate reach-average bed load rates

Existing transport equations represent the available shear stress for a given flow with a single shear stress value, usually the reach-averaged total shear stress estimated as the depth-slope product (Barry et al., 2004; Fernandez Luque and Van Beek, 1976; Meyer-Peter and Müller, 1948; Parker, 1990; Parker et al., 1982; Parker and Klingeman, 1982; Recking, 2013b; Wilcock and Crowe, 2003; Wilcock and Kenworthy, 2002). In other words, most of the available equations assume that this single shear stress value represents the entire shear stress distribution for any given flow level. While this assumption may be appropriate in certain cases, for example in straight reaches with few roughness elements, it is unlikely to represent the hydraulic conditions in complex reaches with variable planforms and roughness characteristics. In our approach, we explicitly account for the local variability in bed surface elevation, channel curvature, and roughness characteristics by including spatially variable estimates of shear stress over a range of hydraulic conditions within the reach, making it more applicable to a wide range of stream types. The main difference between the transport function proposed in equation 8and those typically used when estimating bed load transport rates (e.g., Parker, 1990; Parker et al., 1982; Recking, 2013b; Wilcock and Crowe, 2003) is that equation 8 uses the full distribution of shear stress rather than the reach-averaged shear stress value for a given flow. In practical applications, both approaches require the same input data, specifically, a given discharge, measure of bed roughness,

GSD, and bed surface elevation. While it may be enough for equations using the reach-averaged $\tau$ to define energy gradients using a longitudinal bed profile, our method requires detailed measurements of bed topography to adequately construct a numerical 2D flow model to estimate spatial shear stress distributions. Although acquiring detailed bed surface topography may be restrictive, this method offers an alternative to modern approaches that rely on detailed

field measurements to estimate the $\tau$ applied to the mobile sediment fractions of a given bed. Current flow resistance and shear stress partitioning techniques used in mountain river applications require a characterization of the macro-roughness (Nitsche et al., 2011, 2012) that involve careful field measurements of the diameter, protrusion, concentration, and spacing of boulders (e.g., Monsalve et al., 2016; Yager et al., 2012a), length, slope, and height of steps (Nitsche et al., 2011), and every other source of roughness beside skin friction. Therefore, in general terms,

comparable field effort is required for both modelling of shear stress and estimating of shear stress partition.

We modified a subsurface-based equation to include the spatial distribution of shear stress (i.e., Parker and Klingeman, 1982). Alternatively, we could have chosen to modify the surface-based equation of Parker (1990), also developed using data from Oak Creek, because from a mechanistic point of view it is the bed surface that is in contact with the water whereas the subsurface is not always directly accessed by the flow. However, the fits for the larger number of

345 parameters in the Parker (1990) approach would have been weaker considering the small number of flow events with sufficient information of both the bed load GSD and spatial distribution of shear stress (Table 2) (see also section 4.2). Nonetheless, future improvements of our approach could consider the use of a surface-based equation (e.g., Parker (1990) or Wilcock and Crowe (2003)).

In our equation we used a reach-averaged GSD. Recent studies have shown that including the local $\tau^*$, based

on local shear stress and grain size characteristics, can improve sediment transport predictions in complex mountain rivers (e.g., Monsalve et al., 2016). However, we used a reach-averaged GSD in this study because: i) measuring local grain size distributions (or sediment patches) in a given river is practically complicated for developing a method broadly applicable. This is especially true when trying to delineate submerged sediment patches. ii) the GSD over a reach may vary spatially but the reach-averaged GSD of a given reach is less sensitive to changes in discharge than

the shear stress. Segura and Pitlick (2015) compared the variability of the shear stress distribution and the grain size distribution and found that the shear stress distributions varies more than the GSD, and iii) spatial scale modeling restrictions. 2D models are not able to incorporate the effects of fine scale variability in the surface grain size. Usually the grid cell size in these models are in the order of 20–50 cm. Therefore, even if a detailed grain size distribution were available, fully coupling them within a 2D approach is not yet possible.

**4.2 Alternative formulations for sediment transport prediction using spatial distribution of shear stress**

When calibrating equation 8 we used a total of five flow levels covering a wide range of discharges, from the lower limit (approximately 0.29 $Q_{bf}$) used by Parker and Klingeman (1982) up to bankfull conditions. While conducting the calibration we found an alternative formulation defined by a single equation (instead of a two-part equation like

equation 8 also calibrated for flows above 0.99 m³/s. This equation performed well over a wider range of flows, including those between 0.4 and 0.64 m³/s:

$$W_i^* = 0.38 \left(1 - \frac{1.5}{\phi_i}\right)^{1.5} \qquad \text{, for} \qquad \phi_i \geq 1.5 \qquad \text{(Eq. 16)}$$

The performance of equations 8 and 16 in terms of predicted bed load transport rates and GSD was relatively similar for $Q \geq 0.99$ m³/s (table 3, for simplicity only $D_{50}$ is shown). However, equation 16 also predicted $q_b$ and GSD well for all discharges lower than 0.99 m³/s, with errors below an order of magnitude. When equation 8 is applied to the 0.4 m³/s flow it overestimates the measured bed load rate by 27 times (figure 6). It is important to remark that in the calibration process of equations 8 and 16 the discharge levels of 0.4 and 0.64 m³/s were not used. We presented equation 8 in the results section because it resembles the Parker and Klingeman (1982) equation to which we were comparing (figure 6). However, from practical perspective either formulation could have been used for $Q > 0.99$ m³/s. The ability of equation 16 to accurately capture low flow events is explored in detail in section 4.3.

Table 3: Bed load transport rates ($q_b$) and median grain size estimates ($D_{50}$) using equations 8 and 16.

| $Q$ | $q_{b\_meas}$ | $q_{b\_pred}$ | $q_{b\_pred}$ | $log\left(\frac{q_{b\_pred}}{q_{b\_meas}}\right)$ | $log\left(\frac{q_{b\_pred}}{q_{b\_meas}}\right)$ | $D_{50\_meas}$ | $D_{50\_pred}$ | $D_{50\_pred}$ |
|---|---|---|---|---|---|---|---|---|
| | | Eq. 8 | Eq. 16 | Eq. 8 | Eq. 16 | | Eq. 8 | Eq. 15 |
| (m³/s) | (kg/s) | (kg/s) | (kg/s) | (-) | (-) | (mm) | (mm) | (mm) |
| 0.40 | $1.17 \cdot 10^{-5}$ | $3.11 \cdot 10^{-4}$ | $1.78 \cdot 10^{-6}$ | 1.42 | -0.82 | 4.0 | 13.7 | 3.3 |
| 0.64 | $3.65 \cdot 10^{-4}$ | $3.37 \cdot 10^{-3}$ | $1.56 \cdot 10^{-3}$ | 0.96 | 0.63 | 5.1 | 19.9 | 16.2 |
| 0.99 | $3.01 \cdot 10^{-3}$ | $6.23 \cdot 10^{-3}$ | $3.31 \cdot 10^{-3}$ | 0.32 | 0.04 | 7.7 | 20.0 | 20.0 |
| 1.33 | $1.51 \cdot 10^{-2}$ | $8.18 \cdot 10^{-3}$ | $4.32 \cdot 10^{-3}$ | -0.27 | -0.54 | 13.6 | 19.9 | 19.9 |
| 1.46 | $2.00 \cdot 10^{-2}$ | $9.76 \cdot 10^{-3}$ | $5.37 \cdot 10^{-3}$ | -0.31 | -0.57 | 10.2 | 20.1 | 20.1 |
| 1.91 | $2.8 \cdot 10^{-2}$ | $4.11 \cdot 10^{-2}$ | $3.44 \cdot 10^{-2}$ | 0.17 | 0.09 | 18.8 | 21.3 | 21.3 |
| 3.40 | $3.78 \cdot 10^{-1}$ | $3.76 \cdot 10^{-1}$ | $3.76 \cdot 10^{-1}$ | 0.00 | 0.00 | 30.1 | 22.5 | 22.5 |

**4.3 Comparison between equation 8 and 16 and Parker and Klingeman (1982)**

Not surprisingly the subsurface-based sediment transport equation of Parker and Klingeman (1982) gives accurate estimates of bed load for flow events capable of breaking the pavement in a certain reach, given that the equation was exclusively developed for those conditions. Since we are presenting a new approach for estimating bed load transport rates we compared the performance of equations 8 and 16 to the Parker and Klingeman (1982) equation. First, we studied the accuracy of these three methods for 27 events with flow discharges larger than 1 m³/s (figure 8 a). All approaches had practically an equal performance when predicting these sediment transport events and had estimates within an order of magnitude of error (figure 8 b). The equations of Parker and Klingeman (1982) and equation 8 predicted a total of 16 events (59%) within factor of 2 (between 0.5 to 2 times the measured bed load rate), whereas and equation 16 predicted 14 events within this range (52%). Compared to Parker and Klingeman (1982) equations 8 and 16 under predicted bed load for most of the events but had a slight improvement in terms of the RMSE of the predicted bed load transport rate (figure 8 b).

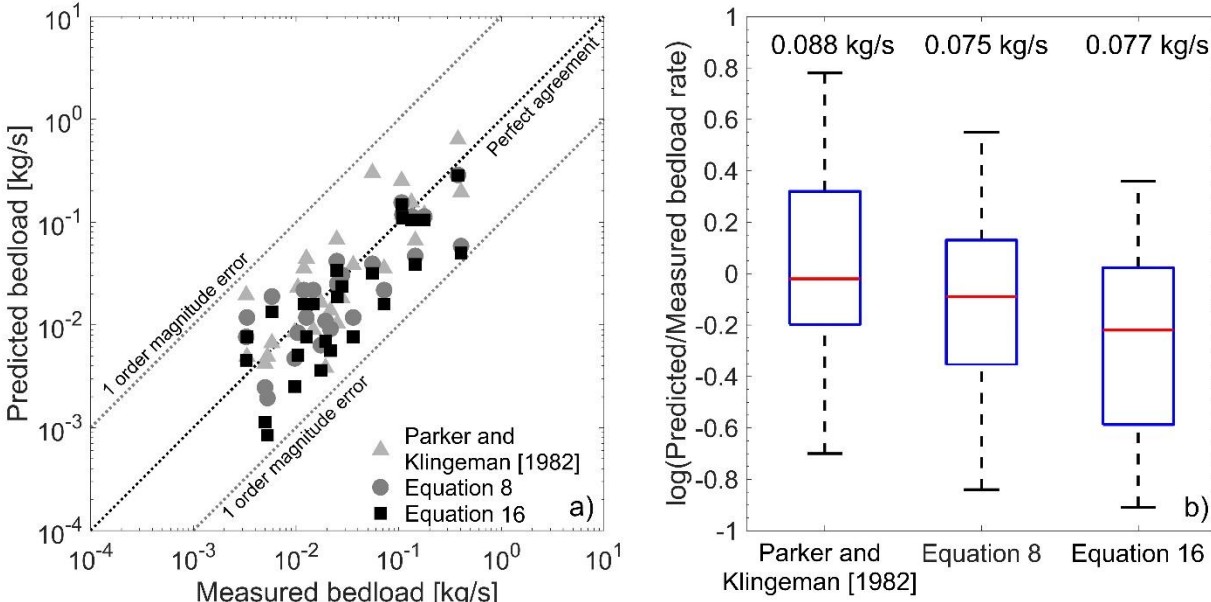

Figure 9: a) Comparison between measured and predicted bed load transport rate using the Parker and Klingeman (1982) equation and equations 8 and 16 In this case, Parker and Klingeman (1982) was applied as proposed in the original publication (i.e., using reach-averaged flow properties). Equations 8 and 16 use spatial distributions of $\tau$ obtained with a Gamma function and $\alpha$ and $\beta$ parameters varying with $Q$. b) The log of the ratio of predicted to measured sediment bed load rate for the three approaches. A value of zero indicates that the measured volume was predicted exactly. The top and bottom of each box are the 25th and 75th percentiles and the middle line inside the box is the median value. Lines extending out of the box correspond to the maximum and minimum predicted bed load ratios. The rate at the top of each box corresponds to the RMSE of the predicted bed load rate.

One limitation of Parker and Klingeman (1982) equation is that it is valid only for $\phi > 0.95$. In practical terms, a value of $\phi = 0.95$ in Oak Creek is close to the already mentioned discharge of 1 m³/s. This relatively high value introduces a practical limitation in the applicability of this method because low discharges are more frequent than high flow events. According to 266 observed sediment transport events in Oak Creek, including the data of Milhous (1973) and measurements collected 1978–1990, the majority of the monitored events (~86%) were at discharges below 1 m³/s. In all these cases (230 events) a bed load transport rate was measured. Using this data set we tested the performance of equations 8 and 16for predicting low and high flow events that vary between 0.01 and 3.4 m³/s. Given that our equations use the distribution of shear stress they, theoretically, should predict sediment transport even at relatively low flows and, by doing so, they would overcome the limitation of the Parker and Klingeman (1982) formulation.

Equations 8 and 16 predicted relatively similar bed load rates for discharges above 0.8 m³/s (figure 9 a). For $Q < 0.8$ m³/s, the equations behave differently. Equation 8 had consistently larger $q_b$ compared to equation 16. The difference between equations 8 and 16 increased as flow discharge decreased and the maximum difference was about 15 times for $Q = 0.01$ m³/s (figure 9 a). We found that 91% of the observed sediment transport events in Oak Creek were predicted within an order magnitude (figure 9 b) with equation 16. In general, equation 8 under predicted $q_b$ while equation 16 over predicted $q_b$. Specifically, equation 8 over predicted bed load for 72% of the discharge events while

equation 16 under predicted bed load for 67% of the discharge events (figure 9 c). These discrepancies were also reflected in the distribution of the ratio of predicted to measured bed load rate values (figure 9 c). Considering the logarithm of this ratio, for equation 8 the 25th, 50th, and 75th percentiles were 0.52, 0.92, and 0.04 while for equation 16 the 25th, 50th, and 75th percentiles were -0.24, 0.14, and -0.58 (figure 9 c). Contrary to Parker and Klingeman (1982), equations 8 and 16 were able to predict $q_b$ at low flows because the lower limit of $\phi = 0.95$ in equation 8 or $\phi = 1.5$ in equation 16 did not correspond to a given discharge (1 m³/s in the case of Parker and Klingeman (1982)). Instead, when using our equations (8 and 16), $\phi$ varies locally with $Q$ such that it captures high values of $\tau$ that occur even at very low flows in small portions of the bed.

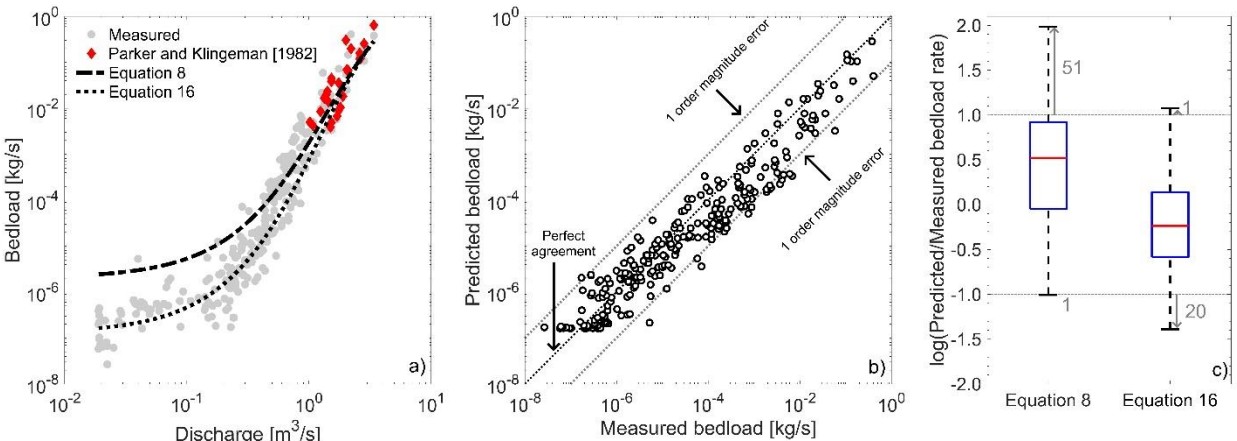

Figure 10: a) Measured and predicted bed load transport rates as a function of discharge. Equations 8 and 16 can be represented as a continuous line because the spatial distributions of $\tau$ and the $\alpha$ and $\beta$ parameters vary with $Q$. Predictions of Parker and Klingeman (1982) were calculated using the reach-averaged shear stress based on Milhous (1973) measurements. Therefore, shear stress does not monotonically increase with larger discharges. b) Measured versus predicted bed load transport rate using equation 16. c) The log of the ratio of predicted to measured sediment bed load rate for equations 8 and 16. Grey arrows extending out of the box correspond to the number of events under- or over-predicted bed load by more than an order magnitude error.

## 4.4 Practical and management implications

The ability of the proposed transport equations 8 and 16 to accurately predict bed load transport rate at a wide range of flows allows our approach to be applied across many different practical scenarios. For small streams like Oak Creek (less than ~10 m in bankfull width) with relatively simple channel geometry and low relative roughness, equations 13–15 can be combined with equation 16 to estimate $q_b$ across a range of flow levels and without a 2D hydraulic model. Equations 13–15 can first be used to estimate the $\tau$ distribution for a given discharge level and then equation 16 can be used with that distribution to estimate $q_b$. Because streams of this type are fairly ubiquitous in modern urban and suburban society, this method can be applied to a range of management situations such as addressing elevated sediment loads caused by urbanization or glacial retreat. For larger streams and rivers, our approach can be utilized in conjunction with the development of a 2D hydraulic model to accurately estimate sediment transport using either equation 8 or 16. In all situations, our approach is an improvement on previous methods in predicting bed load transport for lower flow levels. This is especially important because it allows for practitioners to better predict the

responses of management actions on sediment transport dynamics for these more frequent flow levels. It should be noted that, although our method could be capable of predicting fluxes with better accuracy than previous approaches, all our results are based on measurements in single river reach of Oak Creek. Therefore, we would recommend using this method with caution until it has been further tested in other systems.

**5 Conclusions**

Compared to traditional subsurface sediment transport equations that use reach-averaged properties, the proposed equations were able to accurately predict the observed bed load rates at a wider range of flow levels. The shape of the spatial distribution of shear stress was relatively similar for different discharges and allowed us to characterize it in terms of a Gamma function. Therefore, we were able to extend our results to scenarios were no field measurements were made. Nonetheless, increasing the accuracy in bed load estimates requires additional efforts compared to the most approaches (i.e., reach-averaged equations). Specifically, the method proposed relies on detailed numerical flow modelling and field measurements, which can restrict the applicability in typical practical studies. However, this may not be a limitation for its use. Considering that realistic estimates of flow resistance in gravel-bed rivers require a characterization of the all sources of roughness, including macro-roughness elements, both approaches need similar field effort, which is from a practical point of view, the most time-consuming process. In our method, accurate estimate of bed load transport rates at low flow discharge were possible because we explicitly considered high values of $\tau$, even though they occur in small portions of the bed. Future lines of work should include the extension of surface-based bed load equations and exploring how the shape of the spatial distribution of shear stress varies in other rivers with different geomorphological conditions (e.g., step-pool morphologies, steeper slopes, bed surface patchiness, etc.).

**6 Acknowledgments**

We thank to Dr. Pete Klingeman who led the Oak Creek Sediment Transport facility which collected the database used as the basis for this study. Without Pete, none of this would have been possible. This research was funded by the National Science Foundation (Award No. # 1619700) and the USDA National Institute of Food and Agriculture (McIntire Stennis project OREZ-FERM-876). We also thank Dr. Allison Pfeiffer and Dr. James Pizzuto for their constructive comments that helped improve an earlier version of this manuscript. The authors would like to thank the McDonald Dunn Forest Director, Dr. Stephen Fitzgerald, for logistical support setting our study site. Richard McDonald of the USGS provided very useful suggestions in the hydrodynamic model FaSTMECH.

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
