# Peer review of "A bed load transport equation based on the spatial distribution of shear stress - Oak Creek revisit"

_Earth Surface Dynamics, 2020_

## Referee Comment (RC1) · James Pizzuto (Referee) · 9 Jun 2020

This is an interested, well-conceived and executed study. The authors extend our understanding of bedload transport by developing a model that relies on local shear stress to predict local bedload transport rates, which is designed to replace models that only rely on reach averaged shear stresses.

While the manuscript is a fine contribution as is, there are some interesting observations about the nature of bedload transport and previous transport equations that the authors could emphasize more strongly. This would raise the discussion from a more technical level about how the data were collected and analyzed to include new

scientific knowledge that the author's results lead to. For example, it is perhaps not widely recognized that existing transport equations more or less require the the that the entire shear stress distribution in a reach be scaled by a single parameter, the mean that is estimated by the reach averaged depth-slope product. While this assumption may be true in certain cases, for example in straight reaches with few large roughness elements, it is unlikely to be true for complex reaches with variable planforms and roughness characteristics. The author's results provide a mechanistic explanation for why bedload transport predictions using a single reach-averaged shear stress will be inaccurate for these conditions.

It is also meaningful that the author's equation is parallel to, but "lower than", previous equations. This means that most bedload transport occurs in relatively small, localized areas of the bed where shear stresses are higher than average. While not a new or surprising result, it has rarely been convincingly demonstrated by analyzes from specific field sites, or encoded in a method for computing bedload transport. This is a really interesting and important point that can be derived from the author's results. I would encourage them to make more of it in the manuscript.

Jim Pizzuto Dept. of Earth Sciences University of Delaware

Enumerated below are some minor editorial suggestions, keyed to the text.

1. Line 95, "cobble-gravel". Please cite a source for these terms. For most geologists, "gravel" refers to particles > 2 mm, which is obviously not intended here. 2. Line 100. "riparian…" ZONE? Something missing in this sentence. 3. Figure 1. Please give lat-long for this reach so it can actually be found. A location map should allow readers to find the precise area. 4. Line 143 – "subsurface-based". Can you provide some explanation for choosing the sub-surface based equation? This is inconvenient because the subsurface distribution is much harder to measure, and it is philosophically confusing because the subsurface is not directly accessed by the flow. Justification? 5. Line 189 – grouped, not "group". 6. Line 193. Is the variable Fi defined for the

subsurface or the load? The two are not the same. It must be the former, because the latter is not known until qij is computed. Please clarify. 7. Table 1, mean shear stress - how does these values compare to the depth slope product? It would be interesting, even reassuring, to know. 8. Figure 3. I would like to see a map of the spatial distribution of predicted shear stress in the channel, scaled by the depth-slope product, just to be sure that the model predictions appear reasonable. I understand that this may seem unnecessary, but it would help the reader to better understand the results obtained during an important step of the computations. 9. Figure 4a: What trend would be obtained from the depth slope product and steady uniform 1-D flow? Do we actually need the 2-D model to make this correlation? 10. Figure 4b: should the right y axis label refer to the variable beta actually shown in the graph, rather than theta, which is not shown in the graph? 11. Equation 14 – should the dependent variable be beta, rather than Q, which mysteriously appears on both sides of this equation? 12. Line 260. Is a "complete shear stress distribution" the one that is predicted by the 2-D numerical model? Please be clear about this - it is an important detail. 13. Line 264. "relatively strong" - Please let the reader decide if the agreement is "strong". Just describe the correlation using the statistics. 14. Line 274 – "database". Please explicitly note in the text that these results are presented in Fig. 6. 15. Line 280. This is useful, but a better test would be to try to reproduce data from a completely different site (of course outside the scope of this investigation – just offered as an observation).

---

## Referee Comment (RC2) · Allison Pfeiffer (Referee) · 18 Jun 2020

This paper presents a new approach for predicting river bed load flux, calculating transport rate as a function of the distribution of local shear stress throughout the reach rather than a single reach-averaged shear stress. The authors argue that this new approach is better suited to representing transport at low flow because some portions of the bed remain mobile even when the reach averaged calculations would suggest otherwise.

This paper is exceptionally well written. I found it easy to read and digest—especially impressive given that it's a sediment transport paper! I particularly appreciate the au-

thors approach to representing shear stress within a reach using gamma distributions that change systematically with flow.

While this paper could be published as is, I recommend a few minor edits and additions for completeness:

A. I would have assumed that local $\tau^*$ should be calculated using both the local shear stress AND the local grain size. Surface grain size varies substantially throughout a pool-riffle reach, and I might assume that subsurface grain size would as well. The authors should address this point in the text.

B. The authors don't address the fact that their approach has been applied to only one reach of river. Isn't it possible that another channel would be far less conducive to this approach? (e.g. complicated channel geometry could prevent the use of the gamma distribution technique). I think this can be addressed by adding a few caveat sentences to the end of the paper.

C. The choice of a subsurface (rather than surface) relation seems odd, given the goal of calculating sediment transport for lower flows. Shouldn't a surface relation be inherently better at representing low flow transport? I agree that Parker 1990 is convoluted to implement, but this distinction between a surface and sub-surface relation should be more directly addressed.

Line Notes:

Ln 45- I would replace "measuring" with "predicting"

Figure 3 – While not necessary, it would be nice to see the gamma distribution curves overlain on the histograms.

Figures 4, 8, 9 – Remove boxes around individual sub-plots.
* * *

---

## Author Comment (AC1) · 13 Jul 2020

Dear Dr. Pfeiffer and Dr. Pizzuto

We appreciate your thoughtful review of our article and your kind words about our work. We agree with all your suggestions. We believe that after addressing your suggestion the quality of our paper has significantly improved.

Dr. Pizzuto's comments (Referee #1)

General Comments: We divided your general comment in two parts (GC1 and GC2).

GC1: There are some interesting observations about the nature of bedload transport

and previous transport equations that the authors could emphasize more strongly. This would raise the discussion from a more technical level about how the data were collected and analyzed to include new scientific knowledge that the author's results lead to. For example, it is perhaps not widely recognized that existing transport equations more or less require the that the entire shear stress distribution in a reach be scaled by a single parameter, the mean that is estimated by the reach averaged depth-slope product. While this assumption may be true in certain cases, for example in straight reaches with few large roughness elements, it is unlikely to be true for complex reaches with variable planforms and roughness characteristics.

Reply: The method we present based on a 2D flow modeling explicitly accounts for the variability in bed surface elevation and roughness characteristics. In our approach, the spatial variability of shear stress is included and even those low-frequency values (especially local high shear stress values) play a role in determining the estimated bed load transport rate. In the most classical methods, some information is lost in the averaging process, in particular the local high shear stress values. Although, local low values of shear stress are also not well capture by an average shear stress value they are not as relevant given their low contribution to the sediment transport rate. Based on your suggestion we started the discussion highlighting the assumptions of existing bed load equations that rely on a single shear tress value and the advantages of our approach (L 314–323).

GC2: It is also meaningful that the author's equation is parallel to, but "lower than", previous equations. This means that most bedload transport occurs in relatively small, localized areas of the bed where shear stresses are higher than average. While not a new or surprising result, it has rarely been convincingly demonstrated by analyzes from specific field sites, or encoded in a method for computing bedload transport. This is a really interesting and important point that can be derived from the author's results. I would encourage them to make more of it in the manuscript.

Reply: Thanks for the suggestion. We included text reflecting on the implication of the

consistently lower Wi* of the propose equation (L 234–236). We also included a figure (Figure 8) that illustrates the spatial variability of sediment transport, which depends on local shear stress characteristics. The figure has maps of the predicted local bed load transport rate per unit width (qb) for four flow fractions of bankfull discharge (0.29 – 1.00).

Specific comments:

1. Line 95, "cobble-gravel". Please cite a source for these terms. For most geologists, "gravel" refers to particles > 2 mm, which is obviously not intended here

Reply: We included a reference to the work of Milhous, (1973) and a reference to figure 2, which includes the actual grain size distribution of our site (both surface and subsurface) (L 93).

2. Line 100. "riparian. . ." ZONE? Something missing in this sentence.

Reply: Thanks for catching that typo. We added "area" (L 98).

3. Figure 1. Please give lat-long for this reach so it can actually be found. A location map should allow readers to find the precise area

Reply: Thanks for the suggestion. The figure now includes the coordinates in the legend and an improved map for better orientation.

4. Line 143 – "subsurface-based". Can you provide some explanation for choosing the sub-surface based equation? This is inconvenient because the subsurface distribution is much harder to measure, and it is philosophically confusing because the subsurface is not directly accessed by the flow. Justification?

Reply: The motivation for using a subsurface-based equation is explained in the text (L 153–155). We chose this equation because it was developed from measurements collected in the same reach, it gives accurate estimates of bed load transport, and it is relatively simple to extend for our purposes. This subsurface formulation has fewer

**ESurfD**
parameters (or degrees of freedom) than the surface-based option (Parker, 1990) and was relatively easy to adapt as a formulation that includes the complete shear stress distribution. In addition, we had previous experience using this equation with 2D shear stress (Segura & Pitlick, 2015). It is worth mentioning that we are currently exploring the use of the complete distributions of shear stress with a surface-based bed load transport equation. So far, the results are promising and we hope to submit a paper about this very soon. We believe there is value in showing these results as a progression, similar to Dr. Parker and colleagues: first the subsurface formulation (Parker et al., 1982; Parker & Klingeman, 1982) and then a surface-based equation (Parker, 1990).

5. Line 189 – grouped, not "group".

Reply: Thanks for catching that typo. We changed to "grouped" (L 189).

6. Line 193. Is the variable Fi defined for the subsurface or the load? The two are not the same. It must be the former, because the latter is not known until qij is computed. Please clarify.

Reply: Thanks for the suggestion. We specified that it is for the subsurface grain size distribution (L 193).

7. Table 1, mean shear stress - how does these values compare to the depth slope product? It would be interesting, even reassuring, to know.

Reply: We agree. It is an important metric and critical comparison. We included two extra columns into Table 1 to show the average water depth and the total shear stress calculated as the depth slope product. The reach-averaged shear stress calculated using the depth-slope product was consistently higher than the one estimated from de 2D modelling. In general, it provided values that are 126 to 52 % higher than the mean shears tress value derived from the numerical model. In the text we stated "The predicted $<\tau>$ were 66 to 79 % smaller than the mean shear stress values calculated

based on the depth-slope product (Table 1)" (L 2.3–204).

8. Figure 3. I would like to see a map of the spatial distribution of predicted shear stress in the channel, scaled by the depth-slope product, just to be sure that the model predictions appear reasonable. I understand that this may seem unnecessary, but it would help the reader to better understand the results obtained during an important step of the computations.

Reply: We agree and appreciate your suggestion. We now include a figure (Figure 8) with the predicted local bed load transport rate per unit width (q_b) for different flow levels. As anticipated, most of the bed load occurs in a relatively small area of the bed. We do not include the maps for the predicted local shear stress because these are available in a previous publication (Katz et al., 2018).

9. Figure 4a: What trend would be obtained from the depth slope product and steady uniform 1-D flow? Do we actually need the 2-D model to make this correlation?

Reply: Thank you for the suggestion. Given that the shear stress calculated with the depth-slope product is the most common method used to estimate reach-averaged shear stress it is very informative to make the comparison you suggested. We modified Table 1 and Figure 4a to include this information. The following short explanation was also included in Figure's 4 caption "Total shear stress from depth slope product has a similar trend than $<\tau>$ but it is consistently higher. All our results are based on the 2D models and $\tau\_t$ is shown here only for comparison purposes "

Regarding to the question: Do we actually need the 2-D model to make this correlation? If we use the depth-slope product we would only be able to estimate the reach-averaged shear stress (although a different reach-averaged value than the one obtained using 2D modelling). However, it would not be possible to estimate the parameters of the Gamma function ($\alpha$ and $\beta$) or the spatial distribution of shear stresses within a reach. The 2D model is required to obtain these parameters and the distribution of $\tau$ as function of discharge.

10. Figure 4b: should the right y axis label refer to the variable beta actually shown in the graph, rather than theta, which is not shown in the graph?

Reply: Thanks for catching that typo. We edited the figure and now beta is in the right place.

11. Equation 14 – should the dependent variable be beta, rather than Q, which mysteriously appears on both sides of this equation?

Reply: Thanks for catching that typo. We edited the figure and now beta is in the right place and consistent with Figure 4.

12.Line 260. Is a "complete shear stress distribution" the one that is predicted by the 2-D numerical model? Please be clear about this - it is an important detail.

Reply: Thanks for the suggestion. Now it reads" ... applied over the complete shear stress distribution predicted by the 2D numerical model instead of our formulation" (L 256)

13. Line 264. "relatively strong" - Please let the reader decide if the agreement is "strong". Just describe the correlation using the statistics.

Reply: Thanks for the suggestion. We wrote that paragraph again to reflect the suggestion (L 261–262). We also modified Table 2 to include the Nash-Sutcliffe efficiency index.

14. Line 274 – "database". Please explicitly note in the text that these results are presented in Fig. 6.

Reply: Thanks for the suggestion. We modified the text to accordingly (L 271-272).

15. Line 280. This is useful, but a better test would be to try to reproduce data from a completely different site (of course outside the scope of this investigation – just offered as an observation).

Reply: We appreciate your observation. We are currently developing different approaches to estimate bed load transport rates and GSD based on the spatial distribution of shear stress. We believe that this variability must be explicitly included in the estimation of bed load. Part of our current efforts include extending the application of this method to other rivers. We pointed out the need to test this methodology in other systems at the end of the discussion (L 434–437).

Dr. Pfeiffer's comments (Referee #2)

General Comments: We divided your general comment in three parts (GC1, GC2, and GC3).

GC1: I would have assumed that local $\tau^*$ should be calculated using both the local shear stress AND the local grain size. Surface grain size varies substantially throughout a pool-riffle reach, and I might assume that subsurface grain size would as well. The authors should address this point in the text.

Reply: You raised an interesting issue. There are various motivations for assuming a reach-averaged grain size distribution in this study: a) measuring local grain size distributions (or sediment patches) in a given river is practically complicated for developing a method broadly applicable. This is especially true when trying to delineate submerged sediment patches.. We recognize that the variability in the surface grain size is important, in fact, some years ago, one the authors presented a scheme in which the spatial distribution of GSD (in terms of sediment patches) was explicitly included in sediment transport estimates (Monsalve et al., 2016). In that case, the local $\tau^*$ was calculated using the local shear stress and the local grain size. However, sediment patches are difficult to identify and characterize, mainly because their boundaries can change even during low flow variations (Unpublished data) b) although, the GSD over a reach varies spatially the reach-averaged GSD of a given reach is less sensitive to changes in discharge than the shear stress. In a different study Segura and Pitlick (2015) compared the variability of the shears stress distribution and the grain size distribution and found

that the shear stress distributions varies much more than GSD; and c) two dimensional models are not able to incorporate the effects of fine scale variability in the surface grain size. Consider that the grid cell in two dimensional models is in the order of 20–50 cm. Therefore, even if a detailed grain size distribution were available, coupling them within a 2-dimensional approach is yet to be possible.

GC2: The authors don't address the fact that their approach has been applied to only one reach of river. Isn't it possible that another channel would be far less conducive to this approach? (e.g. complicated channel geometry could prevent the use of the gamma distribution technique). I think this can be addressed by adding a few caveat sentences to the end of the paper.

Reply: You raise an important point. We now acknowledge that further testing is required beyond the reach we study (L 434-437). There are very few systems with the kind of data required to test our approach we can currently working on extending this method to other locations. We are also extending the approach to include a surface-based equation.

GC3: The choice of a subsurface (rather than surface) relation seems odd, given the goal of calculating sediment transport for lower flows. Shouldn't a surface relation be inherently better at representing low flow transport? I agree that Parker 1990 is convoluted to implement, but this distinction between a surface and sub-surface relation should be more directly addressed. Reply: This is an important question also raised by reviewer No. 1. The motivation for using a sub-surface-based approach is because the formulation has fewer parameters (or degrees of freedom) and was relatively easy to adapt as a method that includes the complete shear stress distribution. In the original work of Parker and Klingeman (1982) it was shown that the subsurface-based approach provides accurate estimates of bed load transport. We added a paragraph explaining why we chose the subsurface approach (L 337–343).

We are currently exploring the implications of the shape of the distributions of shear

stress, not only for sediment transport predictions but also from a geomorphological perspective. We are aiming to show our results as a progression, similar to Dr. Parker and colleagues: first the subsurface formulation (Parker et al., 1982; Parker & Klingeman, 1982) and then a surface-based equation (Parker, 1990). At this moment we are working in a new article that uses/modifies a surface bed load transport equation.

Specific comments:

1. Ln 45- I would replace "measuring" with "predicting"

Reply: Thanks for noticing that error. Actually, what we wanted to say was: "Bed load modeling can be a convenient alternative to measuring bed load in the field". It was edit and updated in the text (L 43)

2. Figure 3 – While not necessary, it would be nice to see the gamma distribution curves overlain on the histograms

Reply: Thanks, this is a very good idea. We have edited Figure 3 and included the curves for all discharges.

3. Figures 4, 8, 9 – Remove boxes around individual sub-plots. Reply:

Thanks, this is a very good idea. They have been removed.

References

Katz, S. B., Segura, C., & Warren, D. R. (2018). The influence of channel bed disturbance on benthic Chlorophyll a: A high resolution perspective. Geomorphology, 305, 141–153. https://doi.org/10.1016/j.geomorph.2017.11.010 Milhous, R. T. (1973). Sediment transport in a gravel-bottomed stream - Ph.D. Thesis. Department of Civil Engineering, Oregon State University. Monsalve, A. D., Yager, E. M., Turowski, J. M., & Rickenmann, D. (2016). A probabilistic formulation of bed load transport to include spatial variability of flow and surface grain size distributions. Water Resources Research, 52(5), 3579–3598. https://doi.org/10.1002/2015WR017694 Parker, G.

(1990). Surface-based bedload transport relation for gravel rivers. Journal of Hydraulic Research, 28(4), 417–436. https://doi.org/10.1080/00221689009499058 Parker, G., & Klingeman, P. C. (1982). On why gravel bed streams are paved. Water Resources Research, 18(5), 1409. https://doi.org/10.1029/WR018i005p01409 Parker, G., Klingeman, P. C., & McLean, D. G. (1982). Bedload and size distribution in paved gravel-bed streams. Journal of the Hydraulics Division, 108(4), 544–571. Segura, C., & Pitlick, J. (2015). Coupling fluvial-hydraulic models to predict gravel transport in spatially variable flows. Journal of Geophysical Research: Earth Surface, 120, 834–855. https://doi.org/10.1002/2014JF003302

Please also note the supplement to this comment:
https://esurf.copernicus.org/preprints/esurf-2020-25/esurf-2020-25-AC1-supplement.pdf

---

## Author Response (AR2)

**Dear Dr. Polvi**

Thank you very much for your careful review. We appreciate your thoughtful comments and your kind words about our work. We agree with all your suggestions. In this new version we have addressed all of them resulting in a significantly improved article.

*C1: Make sure that common names for plants are in lowercase except for those containing a proper noun (e.g., Douglas, Oregon). Please correct these instances on L98-100 (i.e., 'big leaf maple', 'white oak', 'Oregon white oak', 'alder', 'black cottonwood' & 'Douglas fir'). Also, if 'white oak' and 'Oregon white oak' refer to the same species, please be consistent and use the same name.*

Reply:

Thanks for the recommendation. We edited the paragraph and now it reads: "… dominated by Douglas fir (*Pseudotsuga menziesii*) and Oregon white oak (*Quercu sp*). In the riparian area, vegetation is dominated by alder (*Alnus sp*), black cottonwood (*Papulus trichocarpa*), and big leaf maple (*Acer macrophyllus*) with lower densities of Douglas fir and Oregon white oak" (L 102–101).

*C2: Although the authors addressed Referee #1's comment regarding Figure 1 by adding lat/long ticks, additional context is needed for international readers to be able to easily locate the study area. Please provide an additional smaller-scale inset map to show where Oregon is located within North America to guide international readers. You should also spell out 'Oregon' in the caption for international readers that are not familiar with US state abbreviations.*

Reply:

Thanks for the suggestion. We have edited figure 1. Now it has a map that shows the state of Oregon within the US and the caption reads Oregon instead of Or.

*C3: Both referees raise the issue that a subsurface-based equation is used rather than surface-based sediment transport equation. Although this study is very valuable in providing more*

*accurate predictions of bedload transport, the use of a subsurface- rather than surface-based equation is the one drawback in this study. As Referee #2 said, "a surface relation [should] be inherently better at representing low flow transport." Although the authors provide an explanation why they chose to use the subsurface-based approach in the revised version because it was simpler to implement, I would like the authors to more directly address this issue in the Introduction. For example, in the sentence on L45-48, the authors imply that the drawbacks of previous equations are that they are based on reach-average shear stress estimates or subsurface grain size information. Since this study addresses (only) one of these previous drawbacks, I would like the authors to state this more explicitly in the Introduction and discuss potential drawbacks in sediment transport estimates with basing predictions on subsurface grain size data in the Discussion.*

Reply:

Thanks for the suggestion. We agree, this topic is really important and should be presented at the introduction and discussion as well. We edited the introduction to explicitly account for this suggestion. At the end of the introduction (L 81–85) we added two new sentences: "Although a surface-based equation (e.g., Parker, 1990) would have been stronger at estimating bed load transport at low flows we adopted here a subsurface-based equation (Parker and Klingeman, 1982) because it has fewer parameters to adjust. A subsurface-based equation also allows considering sand sizes, which are commonly found in the bed load (Clayton and Pitlick, 2007; Hassan and Church, 2001; Lisle, 1995; Mueller et al., 2005; Recking, 2010; Segura and Pitlick, 2015)."

A more detailed explanation regarding the choice of using a subsurface-based equation is also given in the discussion (section 4.1, L 341–348). We included a suggestion for future research directions to test and modify our approach (i.e., modify a surface-based equation).

With respect to the drawbacks mentioned by the editor ("*the authors imply that the drawbacks of previous equations are that they are based on reach-average shear stress estimates or subsurface grain size information*") we would like to point out that the drawback that we identified was related to sediment transport equations using reach-averaged one-dimensional shear stress estimates which does not account for the high spatial variability in $\tau$ throughout a river reach. In L 45– 48

we did not mention the use of a subsurface equation as a drawback. We believe that our study shows that a subsurface based equation can accurately predict bed load sediment transport rates even for very low flows. Nonetheless, we agree with the AE that future developments should be in the direction of using a surface-based equation as it was stated in L 347.

C4: Referee #2 raised an interesting issue, as you state yourself, in her GC1 that a local dimensionless shear stress should be calculated using both the local shear stress in addition to the local grain size. You provide a very nice reply to the referee's comment explaining that it is both very complicated to quantify and less sensitive than changes in shear stress. However, as other readers may have the same concern, you should also provide a shorter version of this discussion in the text (suggested in the end of 4.1).

Reply:

We agree, this is very important and it should be included in the text. Thanks for the suggestion. We included the following text at the end of section 4.1 (L 349–359)

In our equation we used a reach-averaged GSD. Recent studies have shown that including the local $\tau^*$, calculated based on local shear stress and grain size characteristics, can improve sediment transport predictions in complex mountain rivers (e.g., Monsalve et al., 2016). However, we used a reach-averaged GSD in this study because: i) measuring local grain size distributions (or sediment patches) in a given river is practically complicated for developing a method broadly applicable. This is especially true when trying to delineate submerged sediment patches. ii) the GSD over a reach may vary spatially but the reach-averaged GSD of a given reach is less sensitive to changes in discharge than the shear stress. Segura and Pitlick (2015) compared the variability of the shear stress distribution and the grain size distribution and found that the shear stress distributions varies more than the GSD, and iii) spatial scale modeling restrictions. 2D models are not able to incorporate the effects of fine scale variability in the surface grain size. Usually the grid cell size in these models are in the order of 20–50 cm. Therefore, even if a detailed grain size distribution were available, fully coupling them within a 2D approach is not yet possible.

*C5: The wording in the added sentence in L339-342 needs slight editing. Change to either "the small number of flow events" or "the few flow events."*

Reply:

Thanks for the suggestion. We have edited the sentence and now it reads: "… considering the small number of flow events with sufficient information of both the bed load GSD and spatial distribution of shear stress". (L 345-347).

*C6: I appreciate that you addressed Referee #1's GC1 in L314-323, but please provide examples and references of existing bedload equations here using reach-scale shear stress values. You already have these in the Introduction, but please point out a few of the most important equations for the readers here in the Discussion as well.*

Reply:

Thanks for the suggestion. We have included the following references (L 318–320):

[revised manuscript text omitted]